# Genome-Wide Identification and Characterization of RNA/DNA Differences Associated with *Fusarium graminearum* Infection in Wheat

**DOI:** 10.3390/ijms23147982

**Published:** 2022-07-20

**Authors:** Guang Yang, Yan Pan, Qinlong Zhao, Jiaqian Huang, Wenqiu Pan, Licao Cui, Weining Song, Therese Ouellet, Youlian Pan, Xiaojun Nie

**Affiliations:** 1State Key Laboratory of Crop Stress Biology in Arid Areas, College of Agronomy, Northwest A&F University, Xianyang 712100, China; yangg@nwafu.edu.cn (G.Y.); panyan@nwafu.edu.cn (Y.P.); zhaoqinlong1072@163.com (Q.Z.); 15829859379@163.com (J.H.); wenqiu_pan@nwafu.edu.cn (W.P.); sweining2002@yahoo.com (W.S.); 2College of Life Science, Jiangxi Agricultural University, Nanchang 330045, China; cuilicao@jxau.edu.cn; 3Ottawa Research and Development Centre, Agriculture and Agri-Food Canada, 960 Carling Ave., Ottawa, ON K1A 0C6, Canada; ouellettr@yahoo.ca; 4Digital Technologies Research Centre, National Research Council Canada, 1200 Montreal Rd., Ottawa, ON K1A 0R6, Canada

**Keywords:** RNA/DNA differences, RNA editing, fusarium head blight (FHB), editing efficiencies, wheat

## Abstract

RNA/DNA difference (RDD) is a post-transcriptional modification playing a crucial role in regulating diverse biological processes in eukaryotes. Although it has been extensively studied in plant chloroplast and mitochondria genomes, RDDs in plant nuclear genomes are not well studied at present. Here, we investigated the RDDs associated with fusarium head blight (FHB) through a novel method by comparing the RNA-seq data between Fusarium-infected and control samples of four wheat genotypes. A total of 187 high-confidence unique RDDs in 36 genes were identified, representing the first landscape of the FHB-responsive RDD in wheat. The majority (26) of these 36 RDD genes were correlated either positively or negatively with FHB levels. Effects of these RDDs on RNA and protein sequences have been identified, their editing frequency and the expression level of the corresponding genes provided, and the prediction of the effect on the minimum folding free energy of mRNA, miRNA binding, and colocation of RDDs with conserved domains presented. RDDs were predicted to induce modifications in the mRNA and protein structures of the corresponding genes. In two genes, TraesCS1B02G294300 and TraesCS3A02G263900, editing was predicted to enhance their affinity with *tae-miR9661-5p* and *tae-miR9664-3p*, respectively. To our knowledge, this study is the first report of the association between RDD and FHB in wheat; this will contribute to a better understanding of the molecular basis underlying FHB resistance, and potentially lead to novel strategies to improve wheat FHB resistance through epigenetic methods.

## 1. Introduction

Wheat is considered one of the most important staple crops around the world and provides approximately 20% of the global dietary energy [1,2]. The continuously increasing and stable production of wheat is essential for ensuring global food security under the challenge of population booming and global climate change [3,4]. Fusarium head blight (FHB), also called scab, caused mainly by the fungus *Fusarium graminearum*, is one of the most destructive diseases of wheat, resulting in huge losses of yield and imposing great health threats for both humans and livestock due to the production of the toxin deoxynivalenol (DON) and related compounds by the fungus [5,6]. More importantly, FHB has gradually become the major hazard and limitation in wheat production in recent years because of climate changes and the expansion of conservation agriculture [7]. Thus, understanding the mechanisms of FHB resistance and developing breeding strategies to produce FHB-resistant wheat cultivars are crucial to cope with these problems. Extensive studies have been carried out to survey resistant germplasms, map QTLs for FHB resistance, and identify major causal genes, revealing some of the mechanisms of the FHB response in wheat [8,9,10]. The cloning and functional validation of *Fhb1* (syn Qfhs.ndsu-3BS) from cv. Sumai 3 was a great breakthrough; this locus is widely used in breeding practice [11,12]. *Fhb7*, which was horizontally transferred from a fungus to wheat, is also a promising locus for FHB resistance breeding [13]. Additionally, using RNA-seq technology, numerous gene expression profiles and gene co-expression network analyses have been performed; these studies have led to the identification of FHB-responsive genes and the discovery of potential regulators and genes associated with constitutive resistance [14,15].

RNA/DNA difference (RDD), also called RNA editing, is a conserved post-transcriptional process involving nucleotide modification, substitution, deletion, or addition within an RNA molecule [16,17]. Together with alternative splicing, RNA editing provides an indispensable process to enrich the genetic information and diversify the transcriptome of an organism; these two processes play a vital role in growth and development as well as stress tolerance in many organisms [18]. Previous studies found that up to 55% of the genetic information in mature mRNA molecules was inconsistent with the initial DNA sequence in plant chloroplasts and mitochondria [19,20]. RNA editing was firstly identified in the mitochondrial genome of trypanosome in 1986, and now it has been widely reported in many species, including animals, plants, as well as fungi [21,22,23]. In mammals, a common type of RNA editing is the deamination of adenosine (A) to inosine (I), which is mainly mediated by the specific ADAR (adenosine deaminase acting on RNA) family of enzymes [24]. An A to I conversion, independent of the ADAR enzyme, has been identified in fungi [25,26]. In plants that were lacking the ADAR gene family, RNA editing was mainly found in the organellar genomes through bioinformatic prediction and a molecular cloning approach; the process is generally regulated by a pentapeptide repeat (PPR) domain protein family [22,27]. With the advances in high-throughput sequencing, RNA-seq technology provides an efficient, unbiased, and economic way to identify RNA editing on a genome-wide scale. Using this method, a large number of studies have been conducted to study the RNA editome or landscape in human and other model organisms, emphasizing the prevalence and importance of RNA editing [28]. However, the study of RNA editing in plants is lagging behind; the genome-wide identification of RDDs in a plant nuclear genome has only been performed in *Arabidopsis thaliana* up to now [29]. The plant–pathogen system provides an ideal model to identify RNA editing targets associated with infection based on RNA-seq methods. When comparing the transcriptome sequences of pathogen-treated samples and the counterpart control samples of a given genotype, genotype-specific polymorphisms can be excluded, ensuring the accuracy of RDD identification. This is not the case when comparing sequences from treated samples to a reference genome or between genotypes.

Here, we investigated the RDDs in wheat in response to *F. graminearum* infection using the publicly available RNA-seq samples of four wheat genotypes (Nyubai, Wuhan 1, HC374, and Shaw), at 2 and 4 days post-inoculation (dpi), to understand the roles of RDD in response to FHB in wheat. This study identified RDD sites associated with *F. graminearum* infection, to enrich our understanding of the epigenetic mechanisms of the FHB response in wheat, and also pave the way to investigating the RNA editome using RNA-seq in wheat and beyond.

## 2. Results

### 2.1. Identification of RDDs Associated with FHB Using RNA-Seq Data

Based on the RNA-seq data, a total of 137,037 transcripts in 110,777 gene loci were constructed from the four wheat genotypes, which covered more than 99% of the IWGSC V1.1 reference genome (Figure 1A). A total of 16,399 putative RDD sites associated with FHB were identified using steps 1–3 of the RDD identification method (see Section 4 for more details). There were more RDD sites identified at 4 dpi than at 2 dpi in all three resistant genotypes (Figure 1B), while fewer RDD sites were identified at 4 dpi than at 2 dpi in the sensitive genotype Shaw. Of the editing modifications observed, 5222 RDD sites (36.13%) were of the transition type, with C to T substitutions accounting for 15.05%, and A to G for 21.08%, respectively, representing two highly abundant canonical RDD types [30]. Among the transversions, T to A (17.26%) and C to A (16.73%) substitutions were the most abundant, while T to G and A to C were the lowest ones, with values of 0.92% and 0.90%, respectively. Furthermore, the RDD sites were mainly located between 10 kb upstream and downstream of the transcription start site of the corresponding genes (Figure 1C). Results of annotation showed that most RDD sites (9312) were located in the coding sequence, while 178, 900, and 1926 sites were located in the intron, 3′UTR, and 5′UTR, respectively. A total of 6589 and 2723 RDD sites emerged as nonsynonymous and synonymous variants, accounting for 45.59% and 18.84%, respectively (Figure 1D). In particular, 388 stop-gained editing instances were also identified.

### 2.2. Identification of Putative High-Confidence RDDs Associated with FHB

To obtain the high-confidence FHB-responsive RDDs, genotype-specific RDDs were removed and the remainder were further verified by the IGV tool (see Section 4 for more details). After verification, a total of 187 unique high-confidence FHB-responsive RDDs were identified, located in 36 unique genes (Figure 2A and Appendix A), representing the first landscape of an FHB-responsive RDD dataset in wheat. Of these, four RDDs were found only at 2 dpi and 164 only at 4 dpi; there were 19 RDDs found at both 2 and 4 dpi, in six genes (TraesCS2D02G179300, TraesCS2D02G405500, TraesCS3A02G263900, TraesCS4A02G107600, TraesCS5A02G073800, and TraesCS6A02G119700). There were 43 RDD sites common among the three FHB-resistant genotypes (HC374, Nyubai, Wuhan 1) yet absent in susceptible Shaw, and 146 found in all four genotypes. Two RDDs (chr5A_86181601, chr6A_91424479) were shared by three resistant genotypes at 2 dpi and four genotypes at 4 dpi.

All of the 12 possible substitution editing types were found in the high-confidence RDDs (Figure 2B). The two most abundant editing types were substitutions between C and T (24.1%) and between A and G (26.8%). The majority of these RDDs were located in the protein coding region, with nonsynonymous and synonymous variants accounting for 24.1% and 62.6%, respectively (Figure 2C, Appendix A).

Editing efficiency was defined as the ratio of edited reads to the total mapped reads for each RDD site. The editing efficiency of *F. graminearum*-treated samples was always significantly higher than that of mock samples among the four genotypes. Moreover, the susceptible genotype possessed significantly higher editing efficiency than the resistant genotypes (Figure 2D,E; Appendix A). For example, the editing substitutions at chr3A_487854757 (G > T) and chr3A_487854760 (G > C) of TraesCS3A02G263900, and at chr4A_163818831 (T > C) and chr4A_163818837 (T > C) of TraesCS4A02G126700, all producing synonymous variants, were found in *F. graminearum*-treated samples of all four genotypes only at 4 dpi, and none of them were found in their mock groups, and editing efficiencies were the highest in Shaw and essentially the same among the three resistance genotypes (Appendix A). This was confirmed by mapped reads using the IGV tool (Appendix A). Validation of editing for additional RDDs is provided in Appendix A.

To evaluate the occurrence of similar FHB-responsive RDDs in other wheat genotypes, ten RDD-containing genes (Appendix A) were randomly selected and characterized by Sanger sequencing in samples from the FHB-resistant line R75 and susceptible line S98. The four RDD sites mentioned in the previous paragraph were detected in the cDNA of infected samples but none of them were found in the mock groups of R75 and S98 (Figure 3). No editing was observed in the eight other genes for the RDDs tested.

### 2.3. Expression and Enrichment Analysis of RDD Genes

To further ascertain the role of the FHB-responsive RDDs, we investigated the expression patterns of the 36 high-confidence RDD-containing genes. Results showed that 22 RDD genes were differentially expressed between *F. graminearum*-infected and control samples in at least one genotype (Figure 4; Appendix A). The susceptible Shaw had a higher number of DEGs than any of the resistant genotypes at 2 and 4 dpi. It is interesting that the direction of differential expression was consistent between the genotypes. For example, the heat shock protein gene TraesCS4A02G097900, an ortholog of *OsHsp71.1*, was significantly up-regulated at 2 dpi in Shaw, and at 4 dpi in all of four genotypes. There were four RDDs found at 4dpi in this gene, two nonsynonymous variants (chr4A_109424377, chr4A_109424379) and two synonymous variants (chr4A_109424382, chr4A_109424394), while no RDD was found at 2 dpi in all four genotypes.

To understand the function and regulatory network of these RDD-containing genes, we further performed Pearson correlation analysis between the expression levels of RDD-containing genes (FPKM) and available trait data for the mock and *F. graminearum*-infected samples, including treatment type and duration, percent *F. graminearum* infection, and *F. graminearum* GAPDH mRNA and DON levels, which were available from a previous study [14]. In total, eleven RDD genes were significantly associated with all five phenotypes, indicating that these genes may play important roles in FHB tolerance. Among them, four and seven RDD genes were, respectively, positively and negatively correlated with each trait (Appendix A), of which TraesCS2D02G405500 (ortholog to *Osrubi3*) had the strongest positive correlation, with the average correlation coefficient of 0.83, and TraesCS6A02G119800 (ortholog to *OsEXO70G1*) had the strongest negative correlation, with the average correlation coefficient of −0.76.

GO enrichment analysis of the 36 RDD-containing genes found that the majority of them (77.8%, 28 genes) were associated with the cytosol (GO:0005829, 4.36 × 10^−23^), including 21 genes involved in functions associated with mRNA and ubiquitin protein ligase binding (GO:0003729, 2.82 × 10^−21^; GO:0031625, 7.17 × 10^−18^) and five genes associated with the cellular response to unfolded proteins (GO:0034620, 2.69 × 10^−11^) (Appendix A). Meanwhile, five genes were enriched in the term of defense response to fungus (GO:0050832, 2.20 × 10^−6^). These findings are supported by the KEGG pathway enrichment results. In addition, three calmodulin genes (TraesCS4A02G126700, TraesCS4B02G178200, TraesCS3D02G328300) were identified as involved in the phosphatidylinositol signaling system (ath04070, 8.36 × 10^−5^) and associated with the MAPK signaling pathway (ath04016, 4.25 × 10^−4^) and plant–pathogen interaction (ath04626, 8.37 × 10^−4^).

Interestingly, four RDDs in each of the two homologs of calmodulin-1 (TraesCS4A02G126700 and TraesCS4B02G178200) caused exactly the same four nonsynonymous variations in their coding sequence (Appendix A), indicating that these RDDs may serve a conserved function in the FHB response. Together with the protein coding annotation (Appendix A), the enrichment analyses suggest that most RDD-containing genes are involved in the regulation of key processes in cellular activity, including DNA availability for transcription, and protein synthesis and processing; changes in these processes due to RDD could affect the defense response and resistance to *F. graminearum* infection in wheat.

### 2.4. The Effect of RDDs on RNA Structure

RNA structure is crucial for its function as RNA mainly depends on its local structure to interact with other molecules [31,32]. Therefore, the RDD sites identified in transcripts after *F. graminearum* infection may lead to changes in RNA structure that affect their function. The RNA secondary structure of FHB-responsive RDD-containing genes was predicted by the minimum free energy (MFE) model. RNA secondary structure changes were predicted for 162 RDD sites in 32 genes (Appendix A). Among these sites, the MFE of 75 sites increased after editing, which may increase the instability of RNA, while that of the other 87 sites decreased and possibly enhanced RNA stability. For example, the nine RDD sites of histone H4 gene TraesCS6B02G079200 were predicted to undergo secondary structure changes after editing, including at chr6B_55695035 with a normalized MFE increase of 40.6% and at chr6B_55695233 with a normalized MFE decrease of 70.0%, respectively, ranking as the highest positive and negative predicted changes (Appendix A). Cumulative predicted changes in MFE at the nine RDD sites suggested that the RNA for the histone H4 variant gene would increase in stability after editing. Of note, after *F. graminearum* infection, editing levels at all sites for TraesCS6B02G079200 were highest in the resistant genotype Wuhan 1, although changes in expression level were significant only in the susceptible genotype Shaw. The 15 RDD sites of the histone H3 gene TraesCS7A02G119700 also showed strongly negative cumulative predicted changes in MFE: twelve of the RDDs for this gene were almost completely edited (92–100%) at 4 dpi in Shaw, when the mRNA was significantly upregulated by the infection (Appendix A). In contrast, the 10 RDD sites of another histone H4 variant gene, TraesCS1A02G201300, showed strongly positive cumulative predicted changes in MFE; this mRNA was edited only in the resistant genotypes. Overall, in this dataset, there was no obvious relation between changes in mRNA stability due to editing and changes in mRNA levels.

### 2.5. The Effect of RDDs on Binding Ability and Protein Structure

RDDs due to mRNA sequence editing can impact miRNA–mRNA binding [33]. To better understand the function of RDDs during the wheat response to *F. graminearum* infection, we investigated their effect on miRNA targeting. We identified two miRNAs binding to two genes in areas containing RDD sites (Appendix A). For the majority of miRNAs, minimum binding values were not affected by editing. In contrast, editing generated stable binding conditions for miRNAs *tae-miR9664-3p* and *tae-miR9661-5p* to bind to the *OsRACK1* ortholog TraesCS3A02G263900 and the *HSP70-4* gene TraesCS1B02G294300, respectively.

Analyzing miRNA binding ability to TraesCS3A02G263900 before and after editing revealed that editing to site 1 (chr3A_487854715) created a binding site for *tae-miR9664-3p* (Figure 5A and Appendix A). The RNA secondary structure of this part of the gene was also changed by this editing, with MFE changing from −389.70 to −385.70 kcal/mol (Figure 5B,C), indicating a small reduction in the stability of the RNA structure. Interestingly, the editing efficiency of site 1 was significantly different between the four genotypes, with that in Shaw being the highest (Figure 5D). At 4 dpi, only the expression of TraesCS3A02G263900 was downregulated in Shaw (Figure 5E).

Finally, the impact of editing on the protein products coded by RDD-containing genes was examined by focusing on conserved domains containing RDD sites in these proteins (Appendix A). Forty RDDs causing nonsynonymous changes were associated with conserved protein domains in 19 genes. For TraesCS3A02G263900, two of the RDD sites generated a change in amino acid after editing: chr3A_487854715 (site 1) and chr3A_487854758 (site 5) (Figure 5A and Appendix A). The RDD site 5 of this protein is located within a conserved WD 40 domain (Appendix A).

In TraesCS1D02G258800, an ortholog of *AtTUBB6* belonging to the Tubulin/FtsZ family, five of its nine RDD sites led to a change in amino acid after editing, including four (site 1, chr1D_351247689; site 2, chr1D_351247692; site 3, chr1D_351247703; site 4, chr1D_351247708) that are located in the C-terminal domain of the protein (Figure 6A, Appendix A). RNA secondary and protein 3D structure predictions showed differences after editing. The MFE of the RNA secondary structure changed from −515.30 to −507.10 kcal/mol after editing, indicating a decrease in the stability of the RNA structure (Figure 6B,C). The torsion of the protein 3D structure also changed from −1.81 to −2.20 (Figure 6D,E). Taken together, our results demonstrate that RDDs in mRNA can lead to modifications of RNA and protein folding, which in turn may affect their function and ultimately be involved in modulating the wheat response to FHB.

## 3. Discussion

FHB seriously affects the yield of wheat. Although many achievements have been made in the field of mining disease resistance genes as well as breeding for scab resistance cultivars [13,34,35,36], there is little study on the mechanisms underlying the FHB response and resistance at the post-transcriptional level, especially from the perspective of RDDs. In this study, we identified and manually verified 187 putative RDDs in 36 genes in the wheat nuclear genome associated with FHB. Unlike those RDDs found in organelle genomes that consist mainly of C to T edited substitution, the RDD sites identified here in the wheat nuclear genome are more diverse, encompassing 12 possible editing types, with the transitions C to T and A to G and the transversions T to A and C to A being the most abundant ones. In comparison, U (observed as T in RNA-seq data) to C, A to C, G to U, and A to G changes were the most frequently observed editing types in nuclear genes expressed in *Arabidopsis thaliana* tissues [29,37]. In contrast, the transition RDD types C to U and G to A have been reported to be the most abundant in plant organelles [19,38]. This is the first report on RNA editing associated with FHB and for nuclear genes in wheat; as there is very little information from other plant species, one cannot interpret yet the significance of the changes in frequency for different editing types between wheat and *Arabidopsis*. The RDD sites were mainly located between 10 kb upstream and downstream of the transcription start site of the corresponding RDD genes, with the majority in the genes themselves (coding, 5′, and 3′ UTR), suggesting that the RDD events largely impacted the structure and function of the gene products. Moreover, we also determined that the editing efficiency of the RDDs in susceptible Shaw was significantly higher than that in three resistant genotypes after *F. graminearum* infection. We suspect that editing efficiency may contribute to the FHB response but further study is needed to verify this.

RNA editing is a conserved post-transcriptional modification mechanism that can impact gene expression, RNA binding ability, and protein structure and function [19,39]. The secondary structure of mRNA impacts cell processes mostly through two means: specific secondary structure binding to other molecules and conserved structural protective functional elements [40]. Changes in a protein’s 3D structure caused by RDD events could affect its function, especially when the changes are located within a conserved domain. It has been shown that one RNA editing site in the coding sequence of the nuclear gene *BOSS* affects flowering in tomato [41]. Our results suggested that most FHB-responsive RDD events had a stronger impact on mRNA secondary structures and protein structures than on mRNA levels. In particular, this was demonstrated for tubulin B6 (TraesCS1D02G258800), with changes in RNA and protein structures. The RDD sites in TraesCS1D02G258800 are located in its C-terminal domain. Tubulins form microtubule arrays, which play essential roles in intracellular material transport and in cell division, growth, and elongation; their C-terminal domain is involved in the regulation of the microtubule assembly [42,43,44]. It has also been shown that reactive oxygen species (ROS), which are transiently produced during biotic stresses such as FHB, act as signals in the regulation of the microtubule network [45,46]. More experiments will be required to determine if the RDD events in TraesCS1D02G258800 and its homolog TraesCS1A02G258800, respectively, in wheat subgenomes D and A, affect their function and have an impact on the plant response to FHB.

Some of the RDD events in heat shock *HSP70-4* gene TraesCS1B02G294300 and rice *RACK1* ortholog gene TraesCS3A02G263900 were predicted to produce new binding sites for miRNA *tae-miR9661-5p* and *tae-miR9664-3p*, respectively. Plant miRNA regulates many cell processes, including the response to biotic stresses, mostly through post-transcriptional mechanisms of inhibition [47]. The binding of tae-miR9664-3p to TraesCS3A02G263900 mRNA was possibly facilitated by an increase in MFE and a change in its secondary structure after editing at chr3A_487854715 (G to C). The mRNA levels for this gene declined after infection only in the FHB-susceptible Shaw, in which RDD sites were almost completely edited (90–97%) by 4 dpi. Another RDD site of this gene, chr3A_487854758 (C to G), is associated with a change in amino acid in a WD domain of the gene, which participates in protein–protein interactions with binding partners [48]. In rice, *RACK1* is a regulatory protein that plays a key role in innate immunity, including in the production of ROS and hormone signaling [49]. It is involved in resistance to the rice blast fungus *Magnaporthe grisea* [50]. Interestingly, in rice, miRNA *miR9664*, an ortholog of *tae-miR9664-3p*, has been shown to regulate immunity against the rice blast fungus [51].

As for TraesCS3A02G263900, there was a predicted increase in MFE at the edited RDD site chr1B_512174399 (G to T) in HSP70-4 gene TraesCS1B02G294300, associated with the new miRNA binding site for *tae-miR9661-5p*. Very little is known about this miRNA. Editing at two other RDD sites of TraesCS1B02G294300 produced amino acid changes in the conserved C terminal of the protein, which is associated with substrate binding [52]. Interestingly, only the resistant genotypes are edited at the two RDD sites. *HSP70-4* proteins are part of a conserved family of chaperones that are key in facilitating the folding of novel proteins and/or elimination of misfolded proteins under stress conditions. There are many reports suggesting a positive correlation between HSP70 protein levels and pathogen resistance in plants, with *HSP70-4* being one of the most frequently found in the biotic response [53]. Functional study of the RDD sites associated with TraesCS1B02G294300 and TraesCS3A02G263900 will be required to confirm their involvement in the wheat response to FHB and their effect on wheat resistance to FHB.

The wheat defense response and mechanisms of resistance to FHB are complex and vary between genetically unrelated genotypes [8,35]. When the presence of 22 RDD sites from ten randomly selected genes was tested by Sanger sequencing in two RILs developed from a cross between FHB-susceptible wheat and resistant HYZ, two RDD sites each in *OsRack1* homolog TraesCS3A02G263900 and calmodulin Cam1-3 TraesCS4A02G126700 were detected in *F. graminearum*-infected samples, while the other RDD sites tested were not found. The RILs were genetically distinct from the four genotypes used for the RNA-seq dataset. These initial results suggest that some FHB-responsive RDDs are genotype-specific or present only in a narrow group of genotypes, while others are present in a broader range of genotypes. There could also be some false positive results in RDD identification and PCR amplification, which have been shown to occur when working with complex genome species such as wheat, an allohexaploid species composed of three sub-genomes and more than 80% transposable elements [2,54]. A larger number of RDDs will need to be tested in many more wheat genotypes responding to FHB to better understand the prevalence of such RDDs. Furthermore, the verified high-confidence RDDs enrich the molecular basis underlying FHB resistance, which will pave the way to improvements in wheat FHB resistance through epigenetic methods.

## 4. Materials and Methods

### 4.1. RNA-Seq Data and Read Mapping

A dataset consisting of 48 RNA-seq wheat spike samples was downloaded from the Sequence Read Archive database (SRA, SRP139946, Pan et al. 2018) [14], which comprised four wheat genotypes inoculated with water or *F. graminearum* (strain DAOM233423) in 3 biological replicates and sampled at 2 and 4 dpi. The adaptors, low-quality reads, and unknown nucleotides were filtered from the raw RNA-seq reads using FastQC (version 0.11.8; Andrews et al.; Cambridge UK) and Trimmomatic (version 0.39; Bolger et al.; Golm Germany). The cleaned RNA-seq reads were mapped against the reference genome (IWGSC RefSeq version 1.1; The International Wheat Genome Sequencing Consortium [IWGSC], 2018) using the 2-pass mode of STAR (version 2.7.5c; Dobin et al.; Menlo Park, USA) [55]. The alignments were used for transcript assembly with StringTie (version 1.3.5; Pertea et al.; Baltimore, USA). Furthermore, we quantified the read coverage of each gene by HTSeq (version 0.11.2; Anders et al.; Heidelberg, Germany). Differentially expressed genes were identified using DESeq2 with the adjusted *p*-value less than 0.05 and |fold change| > 2 [56].

### 4.2. Identification of RNA/DNA Difference Sites Related to F. graminearum Infection

The repeated sequences in the bam files generated by STAR (2-pass mode) were marked using the MarkDuplicates tool of Picard (http://picard.sourceforge.net/ (accessed on 28 April 2021)). The reads on the exon were separated by using the SplitNCigarReads tool in GATK (Genome Analysis Toolkit) software (version 4.0; McKenna et al.; Cambridge, MA, USA) [57], and the N error base and the reads in the intron region were removed. The HaplotypeCaller tool in GATK was used to call base polymorphisms with the parameter as follows: --genotype_likelihoods_model ‘SNP’, --stand_call_conf ‘30’, --stand_emit_conf ‘30’ [58]. The lists of base polymorphisms were obtained as a raw gVCF file for each of the 48 samples, and further used for subsequent analysis. To obtain high-confidence RDDs, the raw gVCF files were filtered step by step as follows:(1)Systematic errors of the sequencing platform and software were corrected by the GATK VariantFiltration tool, with the initial filter parameter -filter “FS > 30.0”, -filter “QD < 2.0”;(2)Only RDDs identified in all three biological replicates and mapped by more than 10 reads in each sample with reference read > 2 and edited read > 3 were kept;(3)RDDs detected in both the *F. graminearum*-treated samples and their mock control samples within a genotype were removed to exclude genotype-specific genomic polymorphisms;(4)Approximately 103.8 M single-nucleotide polymorphisms (SNP) from the resequencing data of 414 wheat lines [59] were used to map the obtained RDDs from the above analysis and filter out the putative SNP sites with the same genomic physical position, further removing genomic SNP loci;(5)Finally, the RDDs shared across all four genotypes, or among the three FHB-resistant genotypes (Nyubai, Wuhan 1 and HC374), were retained and the sequence-mapped information for each RDD was further verified by confirming the alignment information of reads with the Integrative Genomics Viewer (IGV) tool.

Through these quality control steps, a stringent set of putative RDDs associated with FHB were obtained. They were annotated with the SnpEff tool (version 3.6; Cingolani et al.; Detroit, USA) [60] using the Chinese Spring wheat RefSeq version 1.1 annotation file downloaded from the Ensemble Plants database (http://plants.ensem-bl.org/index.html (accessed on 25 April 2021)).

To validate these putative RDDs, an FHB-resistant recombinant inbreed line (RIL) (R75) and an FHB-susceptible line (S98) from a wheat RIL population developed by single-seed descent from a cross between the susceptible US wheat variety Wheaton and the resistant Chinese wheat landrace HYZ were used [36]. The plant material growth and *F. graminearum* inoculation were performed following the previously described method by Gong et al. [36]. The *F. graminearum*-inoculated spikelet samples together with their counterpart untreated control samples were collected from 4 to 5 spikes at 2 dpi and three biological replications were performed. The RNA Easy Fast Plant Tissue Kit (Tiangen, Beijing, China) was used to extract total RNA from all samples, and the RT Master Mix Perfect Real-Time Kit (Takara, Dalian, China) was used to synthesize cDNA according to the manufacturer’s instructions. A total of 22 sites in 10 genes were selected based on their orthologs’ functions for Sanger sequencing analysis using the primers listed in Appendix A.

### 4.3. Correlation Analysis between RDD Genes and Traits

The traits data from Pan et al. [14], including treatment type and duration, percent *F. graminearum* infection, *F. graminearum* Glyceraldehyde-3-phosphate dehydrogenase (Fg-GAPDH) mRNA, and DON levels, were used for trait–gene correlation analysis by the Pearson method. A *p*-value ≤ 0.05 was considered as a significant correlation. GO and KEGG enrichment analyses were conducted using KOBAS 3.0 software (version 3.0; Xie et al.; Beijing, China) [61].

### 4.4. RNA Structure Analysis

RNAfold in the Vienna RNA Secondary Structure Package [62] was used to predict the RNA secondary structures of candidate RDD genes before and after editing. In order to compare the RNA structures of different genes, we calculated the normalized free energy of the RNA secondary structure using the same method as described previously [63]. Each candidate sequence was randomly shuffled 100 times to control the base composition before and after editing. Then, the normalized minimum folding free energy (MFE) of each candidate was calculated using RNAfold by
$$\text{z-score}=\frac{mfe_{native}-mfe_{random}}{\sigma }$$
where *mfe_native_*, *mfe_random_*, and *σ* are the MFE in the native sequence, mean, and standard deviation of *MFE* in the 100 random sequences, respectively.

### 4.5. MicroRNA Target Analysis

To determine whether RDDs affected microRNA (miRNA) targeting sites, all of the candidate RDD-containing genes’ transcripts were searched against the published wheat miRNAs in the miRBase using the psRNATarget tool (https://plantgrn.noble.org/psRNATarget/ (accessed on 20 October 2021)) [64], to predict whether they were targeted by a miRNA. The possibility of miRNA targeting in RDD-containing genes was filtered, and the result with the minimum expected value was selected as the optimal prediction.

### 4.6. Protein Domain and Structure Analysis

The PFAM database (http://pfam.xfam.org/ (accessed on 20 October 2021)) was used to predict protein domains via the HMMER v3.3.1 tools (version 3.3.1; Finn et al; Ashburn, USA) [65] with E-value < 1 × 10^−5^. Protein 3D structure was predicted using homology modeling methods in the SWISS-MODEL database (https://swissmodel.expasy.org/ (accessed on 21 October 2021)). We selected the model with the highest agreement with the target protein, using similarity > 30%.

### 4.7. Orthologous Gene Analysis

In order to better ascertain the potential functions of the RDDs genes, all functionally validated orthologous genes in *Arabidopsis thaliana* and *Oryza sativa Japonica Group* were downloaded from TAIR10 (https://www.arabidopsis.org/ (accessed on 25 December 2021)) and Ricedata (https://www.ricedata.cn/gene/ (accessed on 25 December 2021); 2021 release), respectively. Reciprocal best hit BLASTP [66] was used to identify orthologous genes with similarity > 40% and E-value < 1 × 10^−5^.

## 5. Conclusions

To our knowledge, this is the first study to identify RDDs associated with *F. graminearum* infection in wheat at the whole transcriptome level. In total, 187 unique high-confidence RDD sites were identified. The A to G and C to T editing types were found to be the most abundant. Predicted changes in mRNA stability and structure, in protein sequence and structure, and in miRNA binding ability associated with RDD events could modulate the regulation of gene expression and function. This study lays the foundation for further functional studies to reveal the roles of RDDs in the FHB response of wheat, which will enrich the molecular basis underlying FHB resistance and also facilitate FHB resistance improvement through epigenetic methods in wheat and beyond.

## Figures and Tables

**Figure 1 ijms-23-07982-f001:**
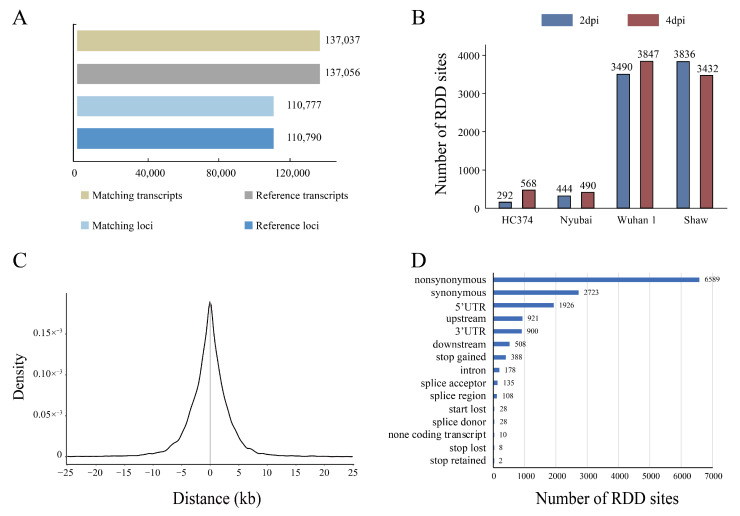
Characterization of RDD sites associated with *F. graminearum* infection using RNA-seq data. (**A**) The number of genes and transcripts assembled from RNA-seq data by mapping to the wheat reference genome IWGSC v1.1. (**B**) The numbers of RDD sites identified in four genotypes at 2 dpi and 4 dpi, respectively. (**C**) The distribution of RDD sites around transcription start site of its corresponding gene. (**D**) Distribution of RDD sites by transcription region. The y axis represents the different types of regions, and the x axis shows the numbers of RDD sites.

**Figure 2 ijms-23-07982-f002:**
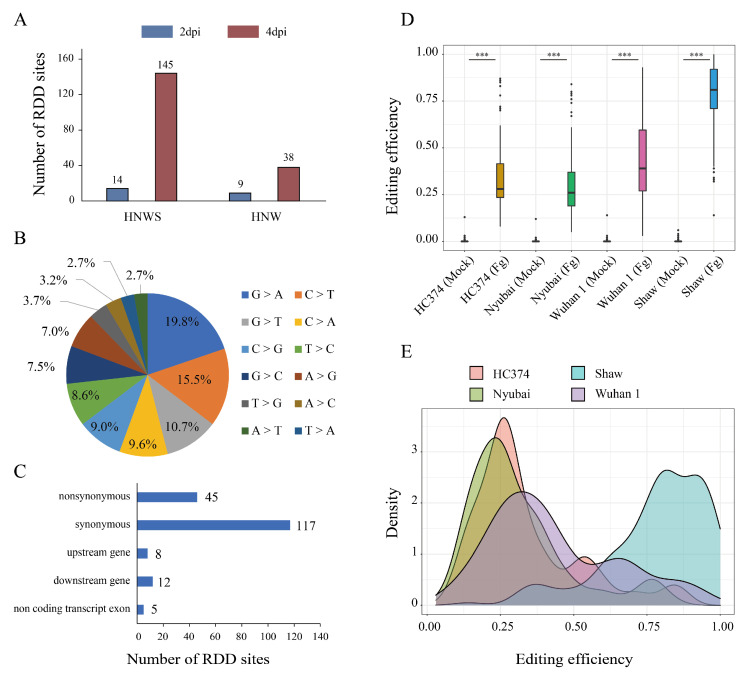
Characteristics of the identified FHB-responsive RDDs. (**A**) Number of RDD sites either common to the four genotypes (HNWS) or common only to the three resistant genotypes (HNW) at 2 and 4 dpi. (**B**) Distribution of the editing types in the high-confidence 187 FHB-responsive RDD sites. (**C**) Distribution of the RDD sites annotated to transcription regions. (**D**) Comparison of editing efficiency between mock and Fg-treated samples among 4 wheat genotypes at 2 dpi and 4 dpi. (**E**) Density distribution of the editing efficiency in each genotype (2 dpi and 4 dpi). Mock: control group; Fg: *F. graminearum*-treated group; ***, Student’s *t*-test *p*-value ≤ 0.001.

**Figure 3 ijms-23-07982-f003:**
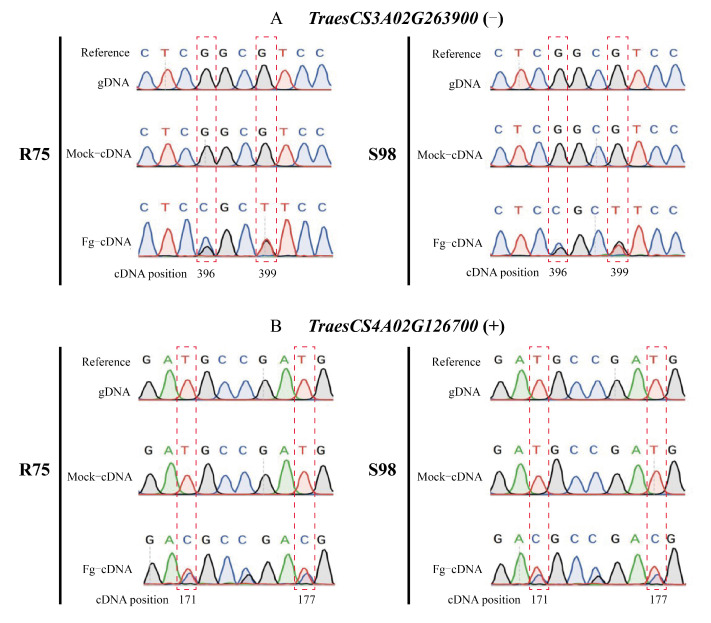
Verification of the RNA–DNA differences in TraesCS3A02G263900 (**A**) and TraesCS4A02G126700 (**B**) through Sanger sequencing. RDDs are framed by a red dotted line. Mock: control group; Fg: *F. graminearum*-treated group.

**Figure 4 ijms-23-07982-f004:**
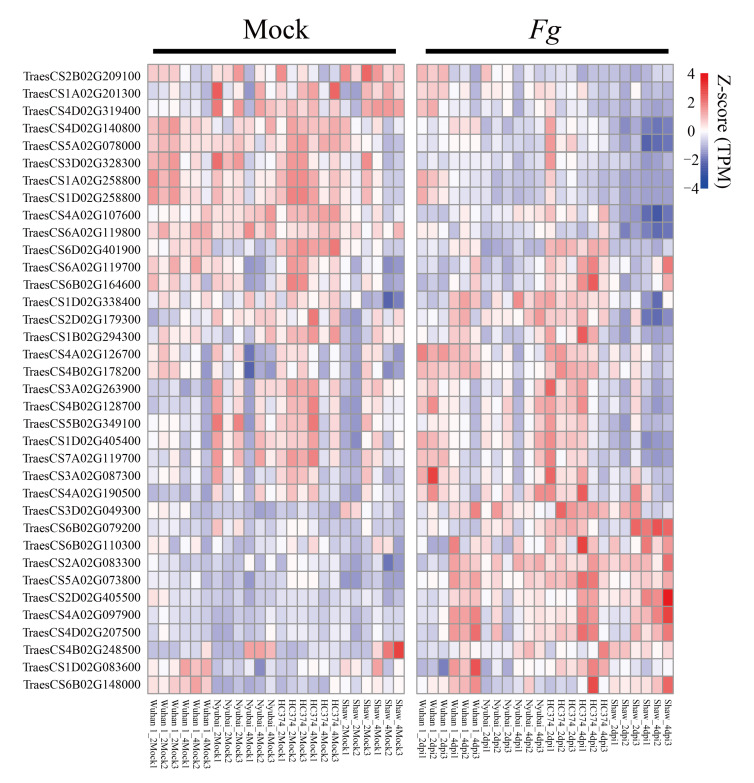
Expression profiles of the 36 RDD genes and their function enrichment. Expression patterns of the 36 RDD genes under mock and Fg-treated conditions among four genotypes at 2 dpi and 4 dpi. Mock: control group; dpi: days post-inoculation.

**Figure 5 ijms-23-07982-f005:**
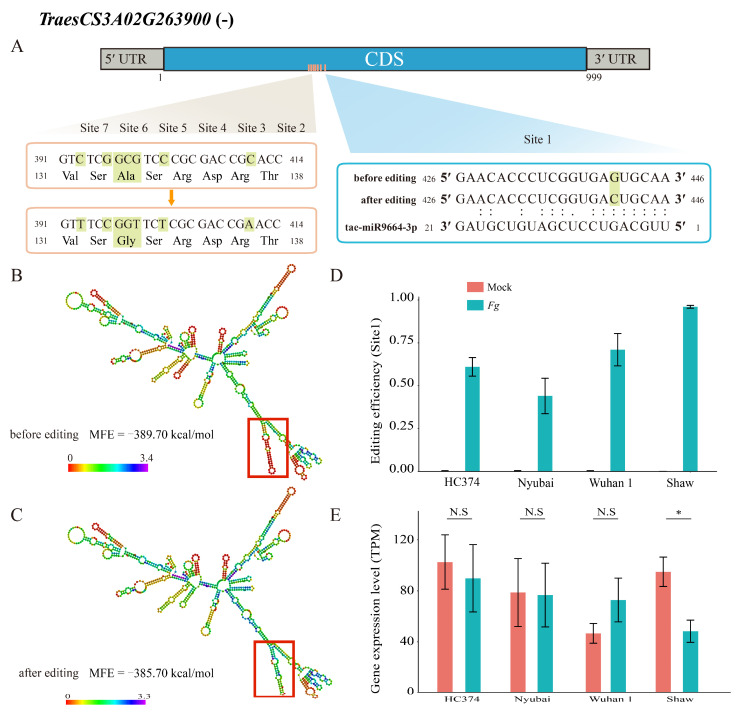
The effect of RNA–DNA differences on the miRNA targeting and RNA 2D structure in TraesCS3A02G263900. (**A**) Seven RDDs were identified in the coding region of TraesCS3A02G263900, of which sites 4 and 5 (chr3A_487854757; 5: chr3A_487854758) were in the same amino acid. Editing in site 1 (chr3A_48785415) generated a miRNA binding site; (**B**) RNA secondary structure of TraesCS3A02G263900 before site 1 editing; (**C**) RNA secondary structure of TraesCS3A02G263900 after site 1 editing; (**D**) editing efficiency of site 1 in four genotypes at 4 dpi; (**E**) expression levels of TraesCS1B02G263900 in four genotypes at 4 dpi. *, Student’ s *t*-test *p*-value < 0.05; N.S, not significant.

**Figure 6 ijms-23-07982-f006:**
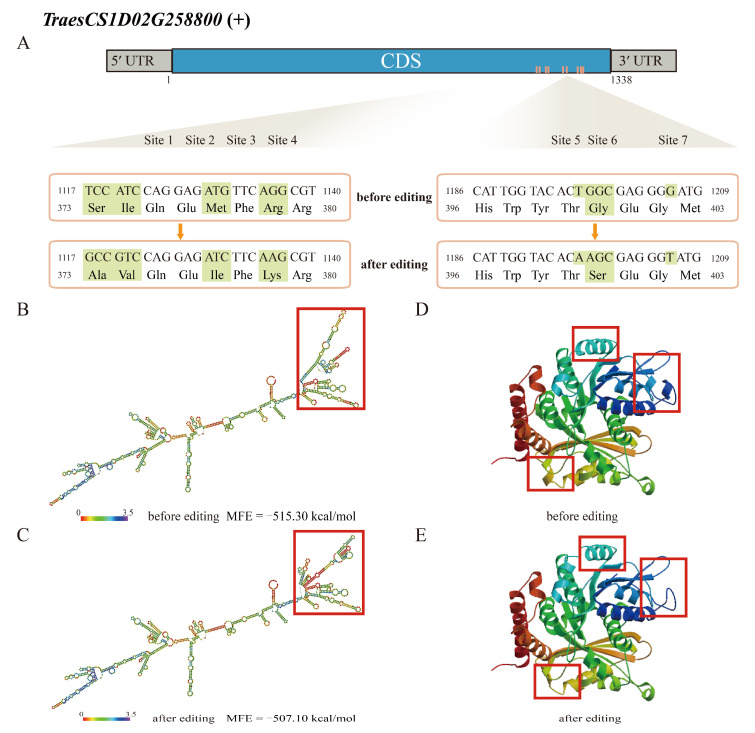
The effects of RNA/DNA differences on the mRNA 2D structure and protein 3D structure of TraesCS1D02G258800. (**A**) Seven of nine RDDs in TraesCS1D02G258800 are shown in the coding region, of which sites 1, 2, 3, 4, and 6 produce an amino acid change; (**B**,**C**) RNA secondary structure of TraesCS1D02G258800 before (**B**) and after (**C**) editing; (**D**,**E**) protein 3D structure of TraesCS1D02G258800 before (**D**) and after (**E**) editing.

## Data Availability

The data that support the findings of this study are available in the Appendix A of this article.

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
