# Peer review of "Genome-Wide Identification and Characterization of RNA/DNA Differences Associated with Fusarium graminearum Infection in Wheat"

_ijms, 2022, doi:10.3390/ijms23147982_

Round 1
Reviewer 1 Report
Dear authors,
Genome-Wide Identification and Characterization of RNA-DNA Differences associated with Fusarium graminearum infection in wheat is well-written manuscript and the topic is relevant and interesting. Authors attempted to understand the RDD associated with fusarium head blight (FHB) by comparing the RNA-seq data between Fusarium-infected and control samples of 4 wheat genotypes. Authors found major significant findings. Overall this shows the close – association between FHB and RDD in wheat. There remain some minor and authors should consider the comments useful for further revision of the manuscript.
Major comments:
This study shows a thorough study with lot of data information. However, one of the problems with this study is authors failed to perform experiments that shows a possible correlation between gene expression and protein expression. It is important to include protein expression studies which further support the RNA seq studies.
Minor comments:
Figure 1 : Figure 1 seem very blurry- the letters are not readable
Discussion section needs to be expanded
Results section needs more clarity
Line 43- more seriously needs to be changed to more importantly.
Author Response
Point 1: This study shows a thorough study with lot of data information. However, one of the problems with this study is authors failed to perform experiments that shows a possible correlation between gene expression and protein expression. It is important to include protein expression studies which further support the RNA seq studies.
Response 1: We appreciated the reviewer's constructive and insightful comment. We agreed that it is important to perform protein expression studies to further support the RNA-seq studies. This study focused on identify the FHB-responsive RDDs through comparison of the RNA-seq data from Fg-treated and mock samples and then the effects of RDDs on gene expression, RNA structure and RNA composition were investigated to get some insights on the biological function of RDDs and also to identify the key candidate genes involving in FHB response and tolerance in wheat from the perspective of epigenetic approach. Thus, we just validated some selected RDD sites based on RT-PCR, without western blotting experiment to investigate protein expression. Actually, our final purpose is to clarify the molecular mechanism of RDD function on FHB response and tolerance. We are now performing the functional studies of some candidates. We appreciated the valuable suggestion and will follow it to perform related experiments in the future study.
Point 2: Figure 1: Figure 1 seem very blurry- the letters are not readable.
Response 2: We are grateful to reviewer for pointing out this sloppiness. We have replaced all Figures of the manuscript with high resolution.
Point 3: Discussion section needs to be expanded.
Response 3: We appreciated the reviewer’s constructive comment. We have expanded the Discussion section in the revised manuscript.
Point 4: Results section needs more clarity.
Response 4: We appreciated the reviewer’s insightful comment. We have further clarified the Results section in the revised manuscript.
Point 5: Line 43- more seriously needs to be changed to more importantly.
Response 5: We appreciated the reviewer’s suggestions. We have fixed it as suggested.
Reviewer 2 Report
Authors have investigated the RDD associated with fusarium head blight (FHB) by comparing the RNA-seq data between Fusarium-infected and control samples of 4 wheat genotypes. The work is is very rich in results and well discussed. A very thorough analysis was conducted. The work for me can be accepted in the IJMS .
Best regards
Author Response
Point 1: Authors have investigated the RDD associated with fusarium head blight (FHB) by comparing the RNA-seq data between Fusarium-infected and control samples of 4 wheat genotypes. The work is is very rich in results and well discussed. A very thorough analysis was conducted. The work for me can be accepted in the IJMS .
Response 1: We greatly appreciate the reviewer's recognition of our job, as well as professional and insightful comments. Best regards !